# Conformal Data Cleaning: Statistical Guarantees for Data Quality Automation in Tables

## Abstract

Machine Learning (ML) components have become ubiquitous in modern software systems. In practice, there remain major challenges associated with both the translation of research innovations to real-world applications as well as the maintenance of ML components. Many of these challenges, such as high predictive performance, robustness, and ethical concerns, are related to data quality and, in particular, to the lack of automation in data pipelines upstream of ML components. While there are many approaches developed for automating data quality monitoring and improvement, it remains an open research question to what extent data cleaning can be automated. Many of the solutions proposed are tailored to specific use cases or application scenarios, require manual heuristics, or cannot be applied to heterogeneous data sets. More importantly, most approaches do not lend themselves easily to full automation.

Here, we propose a novel cleaning approach, *Conformal Data Cleaning* (CDC), combining an application-agnostic ML-based data cleaning approach with conformal prediction (CP). CP is a model-agnostic and distribution-free method to calibrate ML models that give statistical guarantees on their performance, allowing CDC to automatically identify and fix data errors in single cells of heterogeneous tabular data. We demonstrate in extensive empirical evaluations that the proposed approach improves downstream ML tasks in the majority of our experiments. At the same time, it allows full automation and integration in existing ML pipelines. We believe that CDC has the potential to improve data quality with little to no manual effort for researchers and practitioners and, thereby, contribute to more responsible usage of ML technology. Our code is available on GitHub: *redacted GitHub link.*

## 1 Introduction

Machine Learning (ML) components have become ubiquitous in modern software applications. While researchers have made significant progress in developing better models, much of this innovation is only slowly translated into real-world applications. Research at the intersection of ML and database management systems has identified several potential causes, including the difficulty of maintaining an ML system (Sculley et al., 2015) and challenges related to data quality (Breck et al., 2019; Schelter et al., 2018). While many aspects of modern ML workflows have become simpler thanks to standardized APIs and libraries, data quality control remains one of the most impactful and difficult-to-automate parts (Biessmann et al., 2021), especially during ML deployment.

Here we focus on one of the most common and relevant use cases of ML applications, where we assume that an ML model was trained on clean data and at inference time, the data quality deteriorates, impacting the predictive performance. When implementing ML systems, it is common to measure the data quality and remove erroneous examples, e.g., outlier detection (Zhao et al., 2019). However, in many scenarios, low-quality data points cannot be discarded, and a prediction would be desirable. For this reason, one line of research focuses on training robust ML algorithms that are agnostic to data quality issues. For tabular data, these techniques, e.g., regularization (Kadra et al., 2021) and data augmentation (Machado et al., 2022), have been reported to be not as useful yet as for other data modalities (Borisov et al., 2022). On the other

hand, there is a substantial body of literature in ML and database management communities spotlighting data quality and trying to detect which attributes of the examples are erroneous to enable data cleaning. Those approaches often lack the necessary degree of *automation* to apply them to production systems because they rely on user input, e.g., constraints, rules (Rekatsinas et al., 2017), or cleaning suggestions (Mahdavi & Abedjan, 2020) or only work with specific attribute types or error types (Qahtan et al., 2018).

**Contributions** In this study, we aim at improving automation in cleaning tasks by leveraging statistical dependencies inherent to the data. We propose a novel data cleaning approach, Conformal Data Cleaning (CDC), that detects and cleans erroneous attributes of heterogeneous tabular data by exploiting the dependency structure in a table. We combine ML-based data imputation approaches (Jäger et al., 2021) with conformal prediction (CP) (Vovk et al., 2005). Using ML models allows for exploiting the columns' dependencies but calibrating the outputs of such imputation methods can be challenging. We propose to build on the ideas of conformal prediction to ensure that the cleaning results enjoy strong statistical guarantees about whether an attribute, i.e., cell, is conforming to the training data (given other columns' values). In that sense, CDC is a procedure allowing users to combine a wide range of ML models, outlier detection, or data imputation methods to automatically clean (unseen) test data. For this reason, we benchmark CDC on 18 heterogeneous tabular data sets and show that it improves downstream ML tasks in the majority of our experiments.

**Structure of this study** In Section 2, we cover related work. Section 3 describes the theoretical foundation and the main idea of CDC. Section 4 describes the implementation of your experimental study, and Section 5 describes its results, which are discussed in Section 6. Finally, we conclude in Section 7 and sketch ideas for future work.

## 2 Related Work

The term *data cleaning* has different meanings depending on the research area. Here, we briefly describe the differences and highlight how these relate to and influence our work. We focus on tabular data and do not consider methods for other data types (e.g., texts or images).

**Outlier detection** Outlier detection (also known as anomaly detection) identifies data points (examples) that differ substantially from others and is well-known in the machine learning (ML) community. Unsupervised ML approaches to predict whether or not a data point is an outlier is extensively studied for machine learning (Ramaswamy et al., 2000; Liu et al., 2008), empirical cumulative distribution function (Li et al., 2020; 2022) and neural networks (Hawkins et al., 2002; Chen et al., 2017; Wang et al., 2020). Many of the best-performing methods (Han et al., 2022) are available through software packages (Zhao et al., 2019; 2021). The idea of outlier detection is to remove outliers, entire data points, from a data set. In many application scenarios, a complementary goal is to detect and potentially clean anomalous attributes of a data point. While standard outlier detection methods can be applied to this task with minor modifications, these methods are trained on the statistical distribution of a single attribute or dimension and neglect statistical dependencies between attributes.

**Cell-based error detection and correction** Cell-based error detection and correction focuses on errors in individual attributes or dimensions of data points. This task is less studied in the statistical learning community than outlier detection. In contrast, there is a large body of literature in the data management community investigating cell-based error detection and correction methods for relational databases, i.e., tabular data. However, these approaches are often specialized for detecting specific error types, e.g., violations of rules, constraints, or functional dependencies (Dallachiesa et al., 2013; Rekatsinas et al., 2017), inconsistencies (Ham, 2013), or (disguised) missing values ("999999" for a phone number) (Qahtan et al., 2018), and rely on user input. Also, in the data management community, ML methods are increasingly being used for data cleaning, employing semi-supervision (Mahdavi et al., 2019), active learning (Neutatz et al., 2019), or self-supervision (Liu et al., 2022) to exploit user input more efficiently. Most correction approaches can tackle all error types but use user input (e.g., Krishnan et al., 2016; Rekatsinas et al., 2017; Mahdavi & Abedjan, 2020). Abdelaal et al. (2023) show in an extensive benchmark that many of these detection and

correction methods can be combined, which is similar to our approach. Since we focus on improving the automation of cleaning tasks, we build up on methods that do not rely on user input.

**Missing data and data imputation**   Missing values are common data quality issues but do not require complex detection mechanisms. However, after detecting erroneous cells in tabular data sets, one can treat error correction as a data imputation problem. The ML community developed many strategies to handle them, which range from discarding rows or columns over column-wise imputing (Rubin, 1976) and imputing depending on other columns' values (Rubin, 1987; Stekhoven & Buhlmann, 2012; Biessmann et al., 2019) to deep generative models (Yoon et al., 2018; Camino et al., 2019; Yoon & Sull, 2020; Nazábal et al., 2020). Here, we leverage recent work in imputation for heterogeneous data as a central component in our cleaning approach.

**Label error**   We presented data quality issues that relate to the data. However, Northcutt et al. (2021a) show that mislabeled examples are common in computer vision, natural language processing, and audio processing benchmarks. Therefore, different methods are available that detect these label errors Pleiss et al. (2020); Northcutt et al. (2021b); Chen et al. (2021). This type of cleaning is important to obtain a trustworthy training data set. It usually requires manual inspection of the cleaning results to ensure responsible usage of the models trained on the presumably clean training data. Another line of work develops approaches, which are robust against label errors, e.g., by adapting the loss Patrini et al. (2017) or using specialized layers Goldberger & Ben-Reuven (2022). In this work, we consider a complementary problem setting where we aim for automated data (not labels) cleaning at inference time.

To summarize, our work differs from the mentioned studies as we aim for an unsupervised (no user-labeled data, constraints, thresholds, or rules required) approach that detects and cleans erroneous cells of tabular data independent of their error type. Further, we focus on an automated process that can be readily integrated into existing ML pipelines.

## 3   Methodology

In this section, we introduce the theoretical foundation of the conformal framework and develop a generic data cleaning procedure. In short, we build on established ML-based data cleaning approaches and combine them with conformal predictors, allowing us to detect erroneous cells of heterogeneous tabular data to clean them. This simple cleaning procedure, detailed in Section 3.2, is generic enough to account for heterogeneous types of data errors and can be readily implemented in standard ML libraries (see Appendix A).

### 3.1   Conformal Predictors

Conformal predictors are uncertainty quantification methods that allow the calculation of statistically rigorous confidence intervals (regression) or sets (classification) from any point estimator. Vovk et al. (2005) originally described transductive conformal predictors and inductive conformal predictors. However, inductive conformal predictors are much more computationally efficient and were quickly adapted for more complex machine learning (ML) applications (e.g., Papadopoulos, 2008; Balasubramanian et al., 2014). In the following, we refer to inductive conformal predictors as conformal predictors (CP) if not stated otherwise.

#### 3.1.1   Conformal Prediction Framework

Assume $\mathcal{D} := \mathcal{X} \times \mathcal{Y} \subset \mathbb{R}^d \times \mathbb{R}$ is the data space for a regression problem. We sample a training dataset $D_{train} := \{X \times Y\}$ and calibration dataset $D_{calib} := \{X_{n \times d} \times Y_n\}$, the two data sets are independent and identically distributed (i.i.d.).

To obtain a conformal predictor we first fit a machine learning regression model $f : \mathbb{R}^d \to \mathbb{R}$ to the training data $D_{train}$ and obtain the predictor $\hat{f}$. Using the trained model we compute *calibration nonconformity scores* $R_{calib} = r_1, ..., r_n \forall r_i \in [0, \infty]$ for the calibration data $D_{calib}$ using a *nonconformity score function*

$S : \mathcal{Y} \times \mathcal{Y} \to R.$

$$\hat{y}_{calib} = \hat{f}(X_{calib})$$
$$R_{calib} = S(\hat{y}_{calib}, y_{calib}) \tag{1}$$

Intuitively, nonconformity scores represent how different one data point $(x_i, y_i)$ is from what the fitted predictor $\hat{f}$ expects it to be $(x_i, \hat{y}_i)$. Since smaller scores are better, the calibration nonconformity scores $R_{calib}$ will be relatively small if $\hat{f}$ represents $D_{train}$ well. In this example, assume $S(\hat{y}, y) = |\hat{y} - y|$, the absolute error between $y_i$ and its label. This nonconformity score function is commonly used for regression problems.

Then, we compute $\hat{q}$, the $k$-th empirical quantile of $R_{calib}$, as follows:

$$k = \frac{\lceil (n+1)(1-\alpha) \rceil}{n}$$
$$\hat{q} = quantile(R_{calib}, k), \tag{2}$$

where $\alpha \in [0, 1]$ is the given significance level.[1]

Lastly, for new and unseen test data $X_{test}$, we need to construct the prediction interval $\mathcal{T}$. Since we are using the absolute errors as nonconformity scores, we compute the prediction intervals as $\mathcal{T}(X_{test}) = \hat{y}_{test} \pm \hat{q}$. Since $\hat{q}$ is based on the calibration nonconformity scores and the given significance level $\alpha$, the conformal framework guarantees that $\mathcal{T}(X_{test})$ contains $y_{test}$ (the true label) with at least probability $1 - \alpha$, or in other words, with a confidence level of $1 - \alpha$. More formally conformal predictors (CPs) ensure that:

$$\mathbb{P}(y_{test} \in \mathcal{T}(X_{test})) \geq 1 - \alpha. \tag{3}$$

If the model $\hat{f}$ fits the data $D_{train}$ well, the prediction intervals $\mathcal{T}$ will be narrow. However, if $\hat{f}$ performs poorly, the prediction intervals will be broader to satisfy Statement (3). This property is known as *marginal coverage* (Lei & Wasserman, 2014; Lei et al., 2018).

To summarize, by applying the conformal prediction framework, the model predicts intervals or sets that statistically ensure to satisfy Statement (3), independent of the model's choice, its predictive performance, or the data's distribution[2].

### 3.1.2 Conformal Quantile Regression

The marginal coverage property has one major drawback: it does not guarantee good prediction intervals (Lei & Wasserman, 2014) because they have constant widths and can not vary depending on $x$. Besides other approaches, *Conformal Quantile Regression (CQR)* was developed to address this shortcoming (Romano et al., 2019).

The main idea is to fit $f : \mathbb{R} \to \mathbb{R}^2$ to $D_{train}$'s *lower* $q_{\alpha_{lo}}$ and *upper* $q_{\alpha_{up}}$ empirical quantiles to obtain $\hat{f}$, where $q_{\alpha_{lo}} = \alpha/2$ and $q_{\alpha_{up}} = 1 - \alpha/2$. Using the nonconformity score function $S(\hat{y}, y) = max(\hat{y}_{\alpha_{lo}} - y, y - \hat{y}_{\alpha_{up}})$ in the above-described conformal framework and computing the prediction intervals as $\mathcal{T}(X_{test}) = [\hat{y}_{\alpha_{lo}} - \hat{q}, \hat{y}_{\alpha_{up}} + \hat{q}]$, it is possible to calibrate the predicted quantiles to satisfy Statement (3). Since $\hat{y}$ is two-dimensional, $\hat{y}_{\alpha_{lo}}$ represents the predicted lower quantile and $\hat{y}_{\alpha_{up}}$ the upper quantile. For proof or intuitions about the score function, we refer the reader to Romano et al. (2019).

### 3.1.3 Conformal Classification

Transferring the conformal framework to classifiers is straightforward. First, we choose an appropriate nonconformity score function and, second, select a suitable method to construct the prediction sets. For

---

[1]Originally, Vovk et al. (2005) calculated *p-values* for each nonconformity score. However, it has been proven that relying on the fitted residual distribution and $\hat{q}$, as described above, is equivalent (Lei et al., 2018) and commonly used in modern ML applications (Zeni et al., 2020).

[2]There are assumptions for CP, e.g., $|X| > 0$, $|Y| > 1$, and data is *exchangeable*. However, in a typical ML setting, where the data is i.i.d., these are negligible because a sequence of i.i.d. random variables is exchangeable. For more details, we refer the reader to, e.g., Vovk et al. (2005); Zeni et al. (2020).

classification, the label space differs slightly: $\mathcal{Y} \subset \mathbb{N}$. However, many classifiers, especially neural networks using softmax activation, predict probability values for each class. Therefore, classifiers can be seen as $f : \mathbb{R}^d \to [0,1]^{|\mathcal{Y}|}$. It is worthwhile mentioning the worst-case scenario, where these probabilities can be poorly calibrated (Guo et al., 2017). Nevertheless, these are good starting points to define the nonconformity score function. $S(\hat{y}_c) = 1 - \hat{y}_c$ is commonly used, where $y_c$ means the probability score of the true class. The prediction set $\mathcal{T}$ for an unseen test example contains all classes whose nonconformity score does not exceed $\hat{q}$, i.e, $\mathcal{T}(X_{test}) = \{c : S(\hat{y}_{test_c}) < \hat{q}\}$.

Similarly to regression CP, the marginal coverage property does not guarantee good prediction sets: the coverage of the classes can vary heavily. A simple yet powerful extension is the *class-conditioned* conformal classification[3]. In this approach, calibration nonconformity scores are stratified by class, and multiple $\hat{q}^{(c)}$ are calculated. The following equations represent the necessary adaptions, where $\bullet^{(c)}$ represents the subset for class $c$.

$$
\begin{aligned}
R_{calib}^{(c)} &= S\left(\hat{y}_c^{(c)}\right) \\
k^{(c)} &= \frac{\lceil (n^{(c)} + 1)(1 - \alpha)) \rceil}{n^{(c)}} \\
\hat{q}^{(c)} &= quantile\left(R_{calib}^{(c)}, k^{(c)}\right) \\
\mathcal{T}(X_{test}) &= \left\{c : S(\hat{y}_{test_c}) < \hat{q}^{(c)}\right\}
\end{aligned}
\tag{4}
$$

A class-conditioned conformal classifier is guaranteed to satisfy the stronger *class-conditioned* coverage:

$$
\mathbb{P}(y_{test} \in \mathcal{T}(X_{test}) \mid y_{test} = y) \geq 1 - \alpha, \quad \forall y \in \mathcal{Y}.
\tag{5}
$$

In the remainder of this work, we refer to class-conditioned conformal classifiers (CCP) as conformal classifiers.

### 3.2 Data Cleaning with Conformal Predictors

In the following, we demonstrate that the concepts of conformal prediction can help to automate data cleaning routines. We follow an ML-based approach, introduced by van Buuren & Oudshoorn (1999), allowing us to exploit the columns' dependencies and use conformal predictors to detect which cells are erroneous given the information of all other cells in that row. Therefore, during training, we fit an ML model for each column, where all other columns are the model's features, and calibrate its output. During deployment, we (column-wise) test which values are erroneous, meaning which value does not belong to the prediction sets/intervals, and replace them with the underlying ML point prediction.

Formally, let $D^{train}$ be a dataset and *cleaner* our proposed method. Then, *cleaner* fits $d$ models on a subset of the data, where $X_c^{train} = D_{\{1,...,d\}\setminus\{c\}}^{train}$ is the training data and $y_c^{train} = D_c^{train}$ the labels to fit *cleaner*$_c$ to clean column $c \in \{1, ..., d\}$.

**Outlier detection** ML models' predictions are subject to uncertainties, but most model types do not explicitly state them. However, if models predict uncertainties, they are not necessarily well-calibrated (see Section 3.1.3 for an example). This means users can not rely on the model's uncertainty information. Therefore, we use CPs that predict statistically rigorous confidence sets/intervals for a given significance level $\alpha$. For new and unseen test data $D_{n \times d}^{test}$ and significance level, e.g., $\alpha = 0.01$, each of the fitted CP $\widehat{cleaner}_c$ predicts sets/intervals $\mathcal{T}_{i,c}$, where $\forall i \in \{1, ..., n\}$ and $\forall c \in \{1, ..., d\}$, for every test example $i$ of the corresponding column $c$ as follows:

$$
\begin{aligned}
X_{i,c}^{test} &= D_{i,\{1,...,d\}\setminus\{c\}}^{test} \\
\mathcal{T}_{i,c} &= \widehat{cleaner}_c(X_{i,c}^{test})
\end{aligned}
\tag{6}
$$

---

[3]Class-conditioned conformal predictors are also known as *mondrian conformal predictors*. For details and proofs, we refer the reader to Vovk et al. (2005).

If $D_{i,c}^{test}$ is drawn from the same distribution as both the training and calibration data, Statement (3) holds. Hence if $X_{i,c}^{test} \in \mathcal{T}_{i,c}$, we assume that the data point $X_{i,c}^{test}$ is an *inlier*. Otherwise, if the data point $X_{i,c}^{test}$ falls outside the confidence interval $\mathcal{T}_{i,c}$, we assume $D_{i,c}^{test}$ is an outlier. Note that the statistical guarantee in Equation (3) is defined for inliers, not for outliers. One can think of computing a matrix $B_{n \times d}^{test} \subset \{0, 1\}$ according to:

$$B_{i,c}^{test} \begin{cases} 1, & \text{if } D_{i,c}^{test} \notin \mathcal{T}_{i,c} \\ 0, & \text{otherwise} \end{cases} \tag{7}$$

In other words, matrix $B^{test}$ represents outliers of $D^{test}$ as 1, which is valuable for outlier cleaning.

**Outlier cleaning** After detecting outliers, i.e., computing matrix $B^{test}$, it is straightforward to clean them. Having $B^{test}$ representing outliers of $D^{test}$, we can remove the outliers (cell-based) of $D^{test}$ and treat the situation as a missing value problem. However, this requires the application of another potentially independent tool, for example, as shown by Jäger et al. (2021); Nazábal et al. (2020); Yoon et al. (2018); Yoon & Sull (2020); Biessmann et al. (2019).

Since the conformal framework, described in Section 3.1.1, relies on an underlying ML model, we use its predictions to clean outliers. In detail, our calculations for $B^{test}$ differ slightly. Instead of using the values $\{0, 1\}$, we use the *best* prediction of the ML model. Formally, these are minor changes regarding Statement (7). The $\widehat{cleaner}$ CPs return not only a confidence set/interval but also the best prediction for an example used to clean outliers:

$$\mathcal{T}_{i,c}, \hat{y}_{i,c} = \widehat{cleaner}_c(X_{i,c}^{test})$$
$$B_{i,c}^{test} = \begin{cases} \hat{y}_{i,c}, & \text{if } D_{i,c}^{test} \notin \mathcal{T}_{i,c} \\ NAN, & \text{otherwise} \end{cases}, \tag{8}$$

where $NAN$ is a placeholder representing not existing values. Replacing all values of $X^{test}$ with $B^{test}$ if $B_{i,c}^{test} \notin \{NAN\}$ is effectively cleaning $X^{test}$.

## 4 Implementation and Experimental Setup

In this section, we give implementation details of CDC and our baseline, describe our experimental benchmark, and the metrics to compare the results.

### 4.1 Conformal Data Cleaner Implementation

The Conformal Data Cleaner (CDC) implementation, described in Section 3.2, uses our own conformal framework implementation (see Section 3.1.1)[4]. Besides others, it contains classes for CQR (Romano et al., 2019) and CCP based on AutoGluon (Erickson et al., 2020), allowing us to test many ML model types and optimize their hyperparameters (HPO). Since AutoGluon offers a unified API, we do not need to implement the necessary code for each model.

CDC's implementation iterates over each column and, depending on its type, fits a CQR optimizing *pinball loss*[5] or CCP optimizing *F1* metric (two categories) or *macro F1* (more than two categories). Internally, AutoGluon finds the best model from random forests, extremely randomized trees, k-nearest neighbors, linear regression, and a FastAI-based neural network (NN) (Howard & Gugger, 2020). For NNs, it further uses 50 trials (grid search) to optimize their hyperparameters, where we use the default search spaces. Unfortunately, for the other model types, HPO is not implemented, but some predefined hyperparameter settings are tested. We disable model stacking and bagging and expose the best-performing model (without model ensembles) for data cleaning through the API described in Appendix A. Directly afterward, as part of the `fit` interface, we use 1000 data points[6] (not used during training) to calibrate the model.

---

[4]Our conformal prediction framework is publicly available on GitHub: *redacted GitHub link*

[5]*Pinball loss* is the most common metric for quantile regression. See also Section 3.1.2.

[6]Angelopoulos & Bates discussed the effect of the calibration set size and show that 1000 data points works well. However, tabular datasets are typically smaller and calibration set sizes < 1000 could be sufficient. We leave this for future research.

**Data Shifts and empty prediction set** The conformal framework applied to classification problems allows empty set predictions. This typically means that the input data was very different from the training and calibration data, meaning the underlying stationarity assumption was likely violated. In such a scenario, the conformal framework is no longer valid and loses its guarantees – but this is not only true for conformal prediction, as almost all machine learning algorithms implicitly assume stationarity of the data. While some research investigates to what extent algorithms can be made robust against covariate shifts (Sugiyama & Kawanabe, 2012), many of these approaches are designed for special types of algorithms or data distributions (von Bünau et al., 2009). Complementary to this line of work, other researchers have recently proposed several model-agnostic approaches to detect covariate shifts (Rabanser et al., 2018; Schelter et al., 2020) or label shifts (Lipton et al., 2018). In principle, these model-agnostic solutions could be applied as a preprocessing step before applying conformal data cleaning. Empirically, however, we observed that considering cells for which conformal cleaning yields an empty prediction set as inliers and refraining from cleaning these data points resulted in better downstream performance. For this reason, we slightly modify the CDC implementation for categorical columns and check whether the prediction set is not empty before applying Statement (7).

## 4.2 Baseline Implementation

As s baseline cleaning method, we use *PyOD* (Zhao et al., 2019) to detect outliers, more precisely, *ECOD*[7] (Li et al., 2022), which is currently, in many scenarios, one of the best-performing outlier detectors. PyOD combines many algorithms and makes them available through a unified API, which allows us to `fit` and `predict` whether or not a given input example is an outlier. For this reason, we fit iteratively for each column an outlier detector, allowing us to compute cell-wise outlier information. Note, when using column vectors as training data, we are trading information about column dependencies for cell-wise (instead of row-wise) outlier information. After detecting outliers, there are many possibilities to correct these data errors (e.g., Chu et al., 2016; Li et al., 2019). However, a widely used strategy is using the column-wise mean for numerical and mode for categorical columns, which we use for simplicity. We expose the PyOD-based baseline cleaning approach through the cleaning API described in Appendix A.

## 4.3 Cleaning Benchmark

To benchmark the proposed CDC thoroughly, we use openly available datasets from OpenML (Vanschoren et al.) for the three most common task types: regression, binary classification, and multi-class classification. We start from a data imputation benchmark (Jäger et al., 2021) that focuses on the same downstream tasks and remove those that do not fulfill the criteria for tabular datasets formulated by Grinsztajn et al. (2022). We choose datasets with at least $50,000$ cells and prefer fewer columns because CDC fits one model for each column, which can get time-consuming. Appendix B presents the 18 datasets that fulfill our requirements and information about their size and columns.

We split the datasets 80/20 into training and test sets and applied *Jenga*[8] (Schelter et al., 2021) to each of the test sets multiple times to create several corrupted test sets. Jenga can corrupt datasets with realistic error types that model real-world scenarios. However, besides missing values, which are out of the scope of this study, we do not distinguish the error types and refer to them as *outliers*, i.e., observations that differ substantially from others. We use four error types (*swapped values*, *scaling*, *Gaussian noise*, and *categorical shift*) with 1%, 5%, 10%, 30%, and 50% error fractions, which results in 20 corrupted test sets for each data set. Since we do not mix the error types, the actual error fraction after applying Jenga can be smaller than expected, which is caused by the dependency between the error type and the datasets' columns. For example, applying Gaussian noise to a dataset without numerical columns would result in no outliers. Figure 1 gives an overview of our benchmark containing a wide range of error fractions for each downstream task.

---

[7]Unsupervised outlier detection using empirical cumulative distribution functions. For more details, we refer the reader to Li et al. (2022)

[8]"Jenga is an experimentation library that allows data science practitioners and researchers to study the effect of common data corruptions". GitHub: `https://github.com/schelterlabs/jenga`

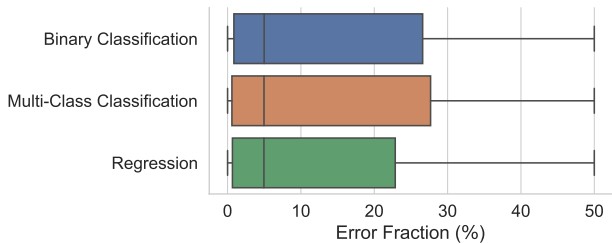

Figure 1: Error distribution. The error fractions are for all downstream tasks similarly distributed. About 22% of the test sets do not have errors, 50% have about 5% or fewer, 75% have about 25% or less, and 25% have between 25% and 50% errors.

### 4.4 Experiments

In ML projects, researchers and practitioners often use high-quality and clean datasets for training and testing. However, in production data quality can degrade, and the (downstream) model's performance drops. Data cleaning methods as part of the ML pipeline have the potential to reduce this effect. To simulate this scenario and empirically benchmark CDC's performance against the PyOD-based baseline, we use the following experimental setup.

We use clean, high-quality training data to fit the downstream model[9] and the two cleaning methods with multiple hyperparameter settings. All experiments are repeated three times to collect information about their robustness. For CDC, this leads to resampling the 1000 calibration data points (see Section 4.1) for each repetition. To compare the two cleaning methods, we use three performance metrics: *baseline performance*, *corrupted performance*, and *cleaned performance*. These measure the downstream performance on the original (clean) test datasets and the erroneous test datasets without and with data cleaning (by each method separately).

**Hyperparameter settings** Both cleaning methods require exactly one hyperparameter, the *confidence level* for CDC, and *contamination* for the PyOD-based baseline cleaner. CDC's confidence level describes how well the cells are required to conform with the training data to be seen as inliers (see Section 3.2). Intuitively, the larger the confidence level, the fewer cells are cleaned. Here, we run six experiments with confidence levels 0.9, 0.99, 0.999, 0.9999, 0.99999, and 0.999999. On the other hand, PyOD's contamination is the proportion of expected outliers[10]. Therefore, the larger, the more cells are cleaned. We run five experiments with contamination equal to 0.1, 0.2, 0.3, 0.4, and 0.499. In the following, we refer to these cleaning method-hyperparameter combinations as *experiment setting*, *experiment*, or *setting*.

### 4.5 Comparison Metrics

Depending on the downstream task, we use *F1* for binary classification, *macro F1* for multi-class classification, and *root mean square error* (*RMSE*) for regression datasets to report their performance measures.

**Normalize performance metrics across datasets** Comparing results across datasets with different difficulties and metrics is not possible. For this reason, we normalize the results for each dataset separately to range between 0 (worst) and 1 (best). This is similar to the distance to the minimum metric used by Grinsztajn et al. (2022) and Feurer et al. (2022). To represent how much cleaning improves (or reduces) the downstream performance, we calculate the downstream performance improvement (corrupted vs. cleaned) relative to the corrupted performance and scale the values for each dataset separately between −1 and 0 for performance degradation and 0 and 1 for performance improvement. This separation is necessary to preserve the information on whether or not the downstream performance improved.

---

[9]We use Jenga, which builds up on scikit-learn's `SGDClassifier` for classification or `SGDRegressor` for regression tasks, pre-processes the columns (scaling, missing value imputation, and one-hot encoding), and optimizes some hyperparameters (grid search).

[10]In real-world scenarios, this can not be known upfront and is a major drawback of this method.

Finally, to report the outlier detection performance, we use the *true positive rate (TPR)*, i.e., probability of detection, and the *false positive rate (FPR)*, i.e., probability of false alarm.

## 5   Results

Before evaluating the results, we average the three repetitions for each experiment, presented in the following. Since there are 120 (corrupted) test sets for each dataset and, therefore, multiple results for each experiment, we use box plots to visualize their distributions. Box plots visually show five summary statistics: minimum, maximum, first quartile, median, and second quartile. We group the results by experiment setting and error fraction to reveal potential trends depending on these variables. Appendix C shows and discusses the results for each downstream task separately.

### 5.1   Outlier Detection

Figure 2 visualizes the outlier detection performance.

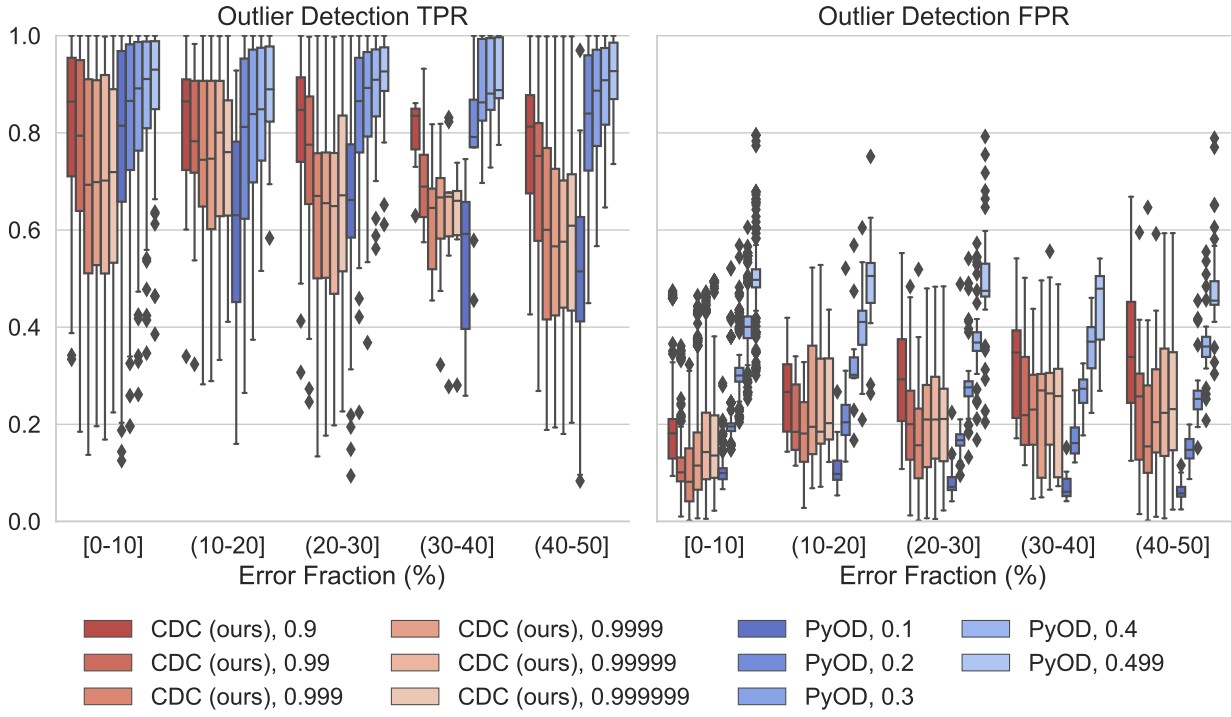

Figure 2:   (*Left*) Outlier Detection TPR (↑) vs. (*Right*) FPR (↓). CDC generally detects fewer outliers than the baseline (left), especially for higher error fractions. CDC with higher confidence levels is more robust regarding its hyperparameter than the baseline, which optimizes the TPR. On the other hand, CDC balances TPR and FPR.

**True positive rate**   In general, CDC's outlier detection TPR (↑) is worse than the baseline but more robust regarding its hyperparameter. Lower confidence levels (0.9 and 0.99) are exceptional and tend to achieve higher TPR. However, the baseline's TPR increases with increasing contamination, which is more pronounced for higher error fractions.

**False positive rate**   The outlier detection FPR (↓) shows similar effects. Compared to the baseline, CDC's results are, especially for high confidence levels, more robust regarding changing hyperparameters.

Increasing error fractions generally degrade CDC's FPR, whereas the baseline increases slightly. However, the baseline's FPR is almost directly defined by its hyperparameter.

**TPR vs. FPR** An optimal cleaner would detect all outliers, i.e., $TPR = 1$, and does not confuse any inliers, i.e., $FPR = 0$. Figure 2 clearly shows that the baseline cleaner focus on outlier detection without minimizing errors. On the other hand, our conformal data cleaning approach focuses on both high TPR and low FPR.

### 5.2 Outlier Cleaning

Figure 3 shows the normalized results of the baseline model applied to cleaned data and the relative improvement over the corrupted performance. The cleaned performance gives insights about which approach works better but does not show whether the performance improved or degraded regarding not using a data cleaning step.

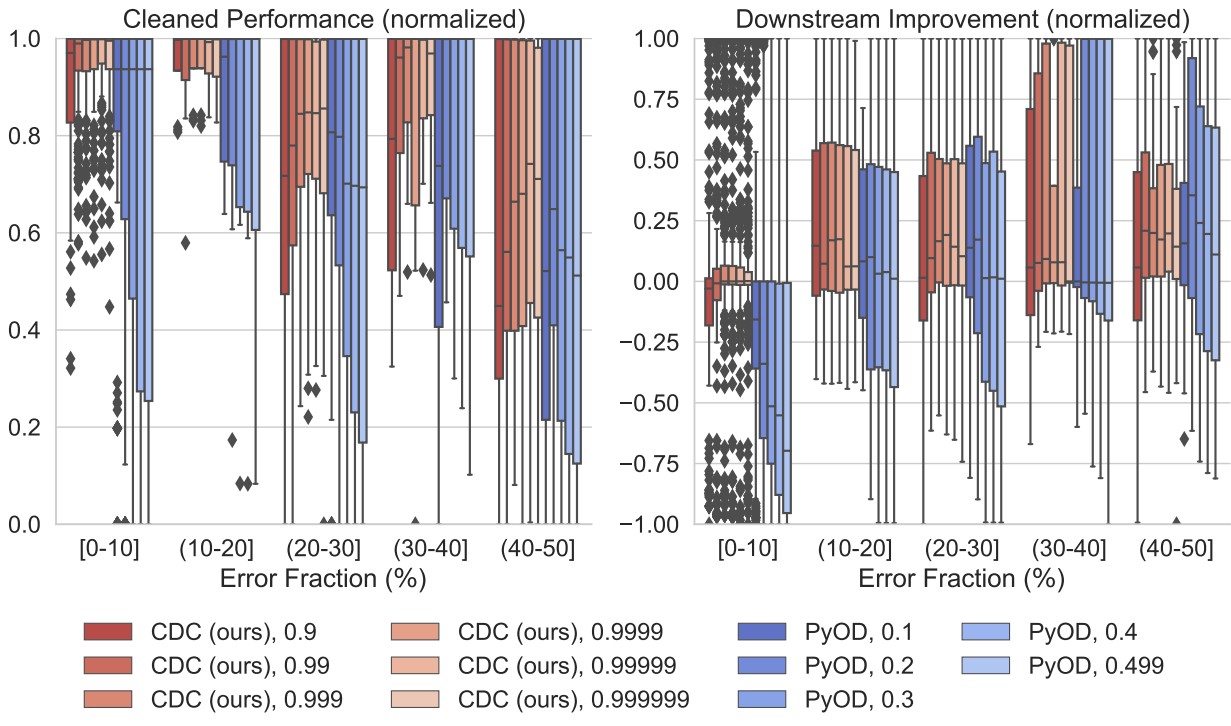

Figure 3: (*Left*) Cleaned performance and (*Right*) downstream improvement (normalized). CDC achieves higher cleaned performance (left) and improves the downstream performance (right) in about 75% of the experiments (first quartile ≥ 0). The baseline's results are more dispersed and among the worst results.

**Cleaned performance** Unsurprisingly, increasing error fractions decrease the downstream performance. However, using CDC performs better than the baseline, especially with higher confidence levels. Furthermore, CDC's boxplots are shorter, meaning the results are less dispersed.

**Cleaned improvement** The baseline's first quartile is always negative. In some cases, the median is close to zero or even negative, meaning that in more than 25% (or 50% for the median) of those experiments, the baseline cleaning approach leads to worse predictive performance in the downstream task. In the range of $[0 - 10]$ percent error fraction (about one-third of the experiments do not have errors, see Figure 1), the baseline leads to degradation in about 75% of the experiments, while CDC achieves improvements in about 50% of the experiments. Further, in about 67% of the experiment settings, at least one hyperparameter for

CDC leads to an improved downstream performance. For the baseline, this is only in 43% of the settings the case. Generally, CDC's results are less dispersed (shorter boxes) and lead to improved predictive performance in the downstream task.

### 5.3 Comparing Best-Performing Results for the Experiments

Figure 4 shows the best-performing experiment settings for CDC and the PyOD-based cleaner. Each colored cross represents one experiment and distinguishes the datasets, while their sizes represent the error fraction. The gray dashed lines visualize the identity. Therefore, marks in the upper left half show experiments where CDC outperforms the baseline.

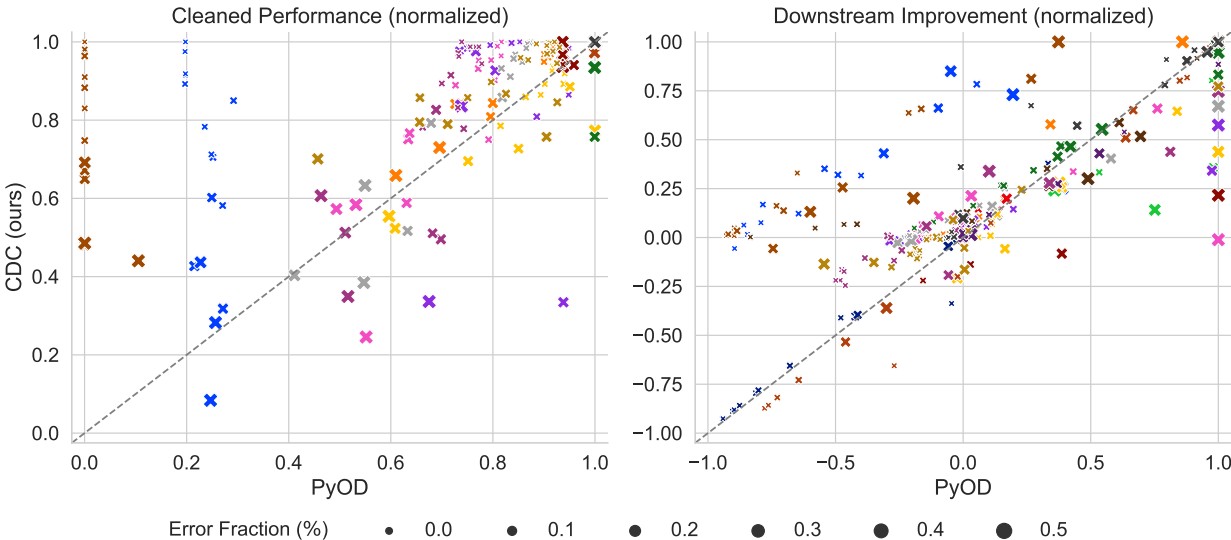

Figure 4: (*Left*) CDC vs. PyOD baseline data cleaning performance and (*Right*) improvement in predictive performance in the downstream task (normalized). Colors distinguish datasets, while cross' sizes represent the error fraction, and diagonal dashed lines visualize the identical performance of the approaches. Experiments above the diagonal lines show that CDC performs better than the baseline, which is the case in 75% of the experiments when comparing the cleaned performance and, respectively, 72.5% for the downstream improvement.

In 75% of the experiments, CDC's cleaned performance (normalized) outperforms the baseline, respectively 72.5% downstream improvement (normalized). Further, in about 67%, CDC leads to improved downstream performance, while the baseline only achieves this in about 42% of the cases. Results for datasets with more errors tend to result in smaller cleaning performance but better downstream improvement. Further, the PyOD-based cleaner shows for many datasets with $\geq 40\%$ error fraction the best results (around 1), while CDC performs worse. This is in line with Figure 3, showing that the baseline's downstream improvement is, in many cases, better than CDC's, represented by larger median and second quartile values.

## 6 Discussion

We investigate the performance of CDC on many datasets and a wide range of artificially created but realistic errors. The results are compared to a PyOD-based cleaning method that does not incorporate column dependencies. In the following, we highlight some of the key findings.

### 6.1 CDC Outperforms PyOD-Based Cleaning

Our experimental results, visualized in Figure 3, show that CDC is superior to the PyOD-based baseline cleaner. CDC's results are, first, less dispersed (smaller boxes in Figure 3), meaning users can expect fewer performance outliers. Second, higher median values for both the cleaned performance and downstream improvements in the majority of experiments. Lastly, the first quartiles are higher (in many cases $\geq 0$) for the downstream improvement, indicating that applying CDC increases the downstream performance in more than 75% of our experiments compared to not cleaning the test data. However, taking Figure 4 into account, this effect also holds when comparing the best-performing model settings for the experiments. These results demonstrate that fully automated cleaning at inference time leads to improvements in the majority of cases. Although, Figure 2 shows that the baseline has better outlier detection TPR. In combination with other results, it is evident that other factors are also important, e.g., smaller FPR and accurate cleaning.

**Influence of error fraction**  Comparing the box-plots median values in Figure 3, higher error fractions ($\geq 30\%$) decrease performance differences between CDC and the baseline, and, eventually, the PyOD-based cleaner outperforms CDC. Figure 4 (right) shows many experiments with larger error fractions, where the baseline outperforms CDC. Multiple errors in a single test example worsen CDC because the underlying ML models suffer from erroneous attributes in (test) samples.

Jäger et al. (2021) show that ML-based approaches using column dependencies for data imputation problems are superior to column-based approaches not using column dependencies. However, they also showed that in the high error fraction regime, ML methods could perform worse. Typically, multiple iterations of data imputation (or cleaning) (Little & Rubin, 2002) improve the results in these scenarios. We leave the implementation and testing of *multiple CDC* for future research.

### 6.2 Influence of CDC's Hyperparameter *Confidence Level*

We describe in Section 3.2 that CDC's hyperparameter *confidence level* directly influences how strong the evidence has to be that a value gets marked as an outlier. Therefore, Figure 2 shows decreasing TPR and FPR with increasing confidence levels. Surprisingly, with *confidence level* $\geq 0.999$, this stagnates and reverses, which is more pronounced for FPR than for TPR. The cleaned performance in Figure 3 (left) shows similarly that increasing confidence levels perform better and finally degrade with *confidence level* $> 0.99999$. These plots show that CDC with $\lim_{confidence\ level \to 1}$ is not necessarily desirable and produces good results for $0.999 \leq confidence\ level \leq 0.99999$.

### 6.3 Tree-Based Models Perform Best in the Majority of Cases

To clean values, CDC fits one ML model for every column. As mentioned in Section 4.1, we use AutoGluon for our experiments to find the best model-hyperparameter combination. In about 46% of these cases, AutoGluon finds a FastAI NN as best performing, in about 31% an extremely randomized tree (XT), and in about 23% a random forest (RF). Given the fact that for FastAI NN, we optimize 50 different hyperparameter settings (random search) but only three for RF and XF each, it is surprising that tree-based models (54%) outperform FastAI NN. However, for data imputation using the same approach, Jäger et al. (2021) already showed that RFs work well, which is in line with a study by Grinsztajn et al. (2022). They provide evidence that tree-based models still outperform neural networks on tabular data. In the future, CDC's performance could be increased by focusing on tree-based models and optimizing their hyperparameters, which we leave for future research.

### 6.4 Limitations

In this study, we use tabular datasets as defined by Grinsztajn et al. (2022) with four to eleven columns (categorical, numerical, and mixed) and about $4,800$ to $89,000$ rows without missing values to benchmark three common downstream tasks: regression, binary classification, and multi-class classification. Thus, our approach and results can not be transferred to other data modalities, such as text- or image-based datasets.

However, since we benchmarked CDC on a wide range of dataset sizes, we assume that it generalizes well to larger or smaller tabular datasets.

We focus on discriminative ML approaches as they are typically used for data cleaning (Abdelaal et al., 2023) and data imputation (Jäger et al., 2021) problems. There is evidence demonstrating that (generative) Neural Networks are often outperformed by tree-based models (Grinsztajn et al., 2022), which is why we focus our empirical comparison on discriminative methods. Applying the ideas of conformal prediction to generative methods is, however, an interesting extension for future research.

As described in Section 4.4, we assume that high-quality training data is available and data errors are tackled at inference time. While this assumption is often violated, we believe that it is a sensible simplification of the problem: First, it allows for substantial improvements in the degree of automation. Our results demonstrate that fully automated cleaning will improve the downstream predictive performance in over 75% of the cases. Second, the curation of training data is often a necessary part of model development cycles in ML. Model development typically involves several iterations, often related to the data preparation stages. Improving data quality and curating the initial training data is an essential step to ensure responsible usage of ML components – the curated data used for training the ML model can usually be used for training the cleaning models without additional curation efforts. Other approaches focus on application scenarios where there is no high-quality training data available to calibrate the cleaning models. Such cleaning systems often circumvent the assumption of high-quality training data by requiring additional user input (e.g., Mahdavi et al., 2019; Mahdavi & Abedjan, 2020; Neutatz et al., 2019; Krishnan et al., 2017; 2016).

## 7 Conclusion and Future Work

In this study, we present how conformalized machine learning models can be used to detect and clean erroneous values of heterogeneous tabular data without requiring user input. We benchmark conformal data cleaning (CDC) on 18 datasets to experimentally compare its performance to a baseline approach. Therefore, we use a wide range of error fractions (5) and error types (4) to create 360 test data sets containing errors.

Our results show that in about 75% of our experiments, using CDC increases the downstream performance while being robust to hyperparameter changes. Importantly these results were obtained without any manual intervention and could be readily used to improve data quality problems at inference time in a variety of application scenarios. In almost all experiments, using CDC outperforms the PyOD-based baseline cleaner. Lastly, we found that in 54% of the cases, tree-based models perform better than FastAI NN, nearest neighbors, or linear regression models. Therefore, we recommend applying CDC with tree-based models and optimizing their hyperparameters more thoroughly. In our experiments, using CDC's confidence level between 0.999 and 0.99999 worked well.

In the future, we plan to test an iterative cleaning approach similar to multiple imputation, which has the potential to further increase CDC's performance, especially with many erroneous values. Further, we plan to investigate the usage of generative ML models.

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

## A  Automated Data Cleaning API

In the ML field, it is, for good reasons, best practice to implement streamlined APIs. Well-known and widely used is the *scikit-learn* API (Pedregosa et al.). Its main components are `estimator`s providing the `fit` method to learn models and `transformer`s offering the `transform` method, which returns a transformed version of the input data (Buitinck et al., 2013). Further, scikit-learn allows the implementation of `pipeline`s to combine different steps, e.g., pre-processing the data (`transformer`) and then fitting/predicting using the ML model (`estimator`).

The cleaning API we propose is integrable into scikit-learn pipelines and consists of four methods:

```
        fit(training_data)
        # Fit conformal predictor for each column of training_data
        return fitted_estimator

        remove_outliers(test_data)
        # Test: test_data's values in prediction sets/intervals
        # If not: remove them
        return data_with_NAN, outlier_mask

        impute(test_data)
        # Impute missing values with CPs best prediction
        return data_without_NAN, imputed_mask

        transform(test_data)
        # Combine 'remove_outliers' and 'impute' methods
        return data_without_NAN_or_outlier, cleaned_mask
```

The methods `remove_outliers` and `impute` are custom interfaces that expose the functionality of detecting and correcting outliers (see Section 3.2). The method `transform` combines these and, therefore, fulfills scikit-learn's transformer abstraction.

## B Datasets

Table 1: Datasets overview. *ID* is the assigned OpenML id, # means the number of, *Cat.* and *Num.* stand for categorical and numerical columns, and *Obs.* means observations, i.e., the number of rows of the tabular dataset.

| ID | Task Type | #Cat. | #Num. | #Obs. | #Cells |
|---|---|---|---|---|---|
| 725 | Binary | 1 | 7 | 8, 192 | 65, 536 |
| 310 | Binary | 1 | 5 | 11, 183 | 67, 098 |
| 1046 | Binary | 1 | 4 | 15, 545 | 77, 725 |
| 823 | Binary | 1 | 7 | 20, 640 | 165, 120 |
| 42493 | Binary | 4 | 3 | 26, 969 | 188, 783 |
| 4135 | Binary | 5 | 4 | 32, 769 | 294, 921 |
| 251 | Binary | 1 | 8 | 39, 366 | 354, 294 |
| 151 | Binary | 2 | 6 | 45, 312 | 362, 496 |
| 40498 | Multi Class | 9 | 2 | 4, 898 | 53, 878 |
| 30 | Multi Class | 1 | 9 | 5, 473 | 54, 730 |
| 198 | Regression | 4 | 2 | 9, 517 | 57, 102 |
| 23515 | Regression | 0 | 6 | 10, 081 | 60, 486 |
| 1199 | Regression | 3 | 6 | 17, 496 | 157, 464 |
| 1193 | Regression | 7 | 2 | 31, 104 | 279, 936 |
| 218 | Regression | 0 | 8 | 22, 784 | 182, 272 |
| 23395 | Regression | 3 | 1 | 89, 640 | 358, 560 |
| 42225 | Regression | 6 | 3 | 53, 940 | 485, 460 |
| 1200 | Regression | 0 | 9 | 59, 049 | 531, 441 |

## C   Results Separated by Downstream Task

To reveal potential relations between downstream tasks and model performance after cleaning, Figure 5 presents the results from Section 5 (Figure 3), additionally grouped by downstream task type. Note, there are eight regression and binary classification but only two multi-class classification datasets (see Table 1).

The normalized cleaned performance for regression datasets (bottom left sub-plot) shows that almost all results are around 1. Because RMSE (the underlying metric) does not have a defined range, which is why, it can, in worse cases, lead to large values dominating the normalization. This typically happens when the error fraction is larger, visualized by diamond markers around $0-0.2$ because many erroneous cells decrease the downstream model's performance. Since the cleaning methods rely on the columns' dependencies, errors also affect their performance.

Besides this, other sub-plots reflect the trends shown in Figure 3 and discussed in Section 5.2. Namely, increasing error fraction decreases the normalized cleaned performance. While CDC typically leads to better results, this performance gap to the baseline reduces with larger error fractions, similar to the downstream improvement. However, for regression with error fractions $> 40\%$, the baseline cleaner outperforms CDC (comparing median values). Generally, the baseline leads to more dispersed results (larger boxes), except for the normalized downstream improvement on regression tasks, where both show exceptionally larger variance.

Not surprisingly, comparing the best-performing settings separately for each downstream task, we do not find a systematic effect of the downstream task in Figure 6. As mentioned in Section 5.3, in 75% of the experiments, CDC's cleaned performance (normalized) outperforms the baseline, respectively 72.5% downstream improvement (normalized). Noteworthy, this trend is also valid for all downstream tasks: multi-class classification 90% and 87.5%, binary classification 78.1% and 75.6%, and regression 68.1% and 65.6%  Further, CDC improves the downstream performance in about 55% of the multi-class classification tasks, where the baseline achieves improvements in only 20% of the experiments. Similarly for binary classification tasks with 63% and 39%, and for regressions tasks with 74% and 50%.

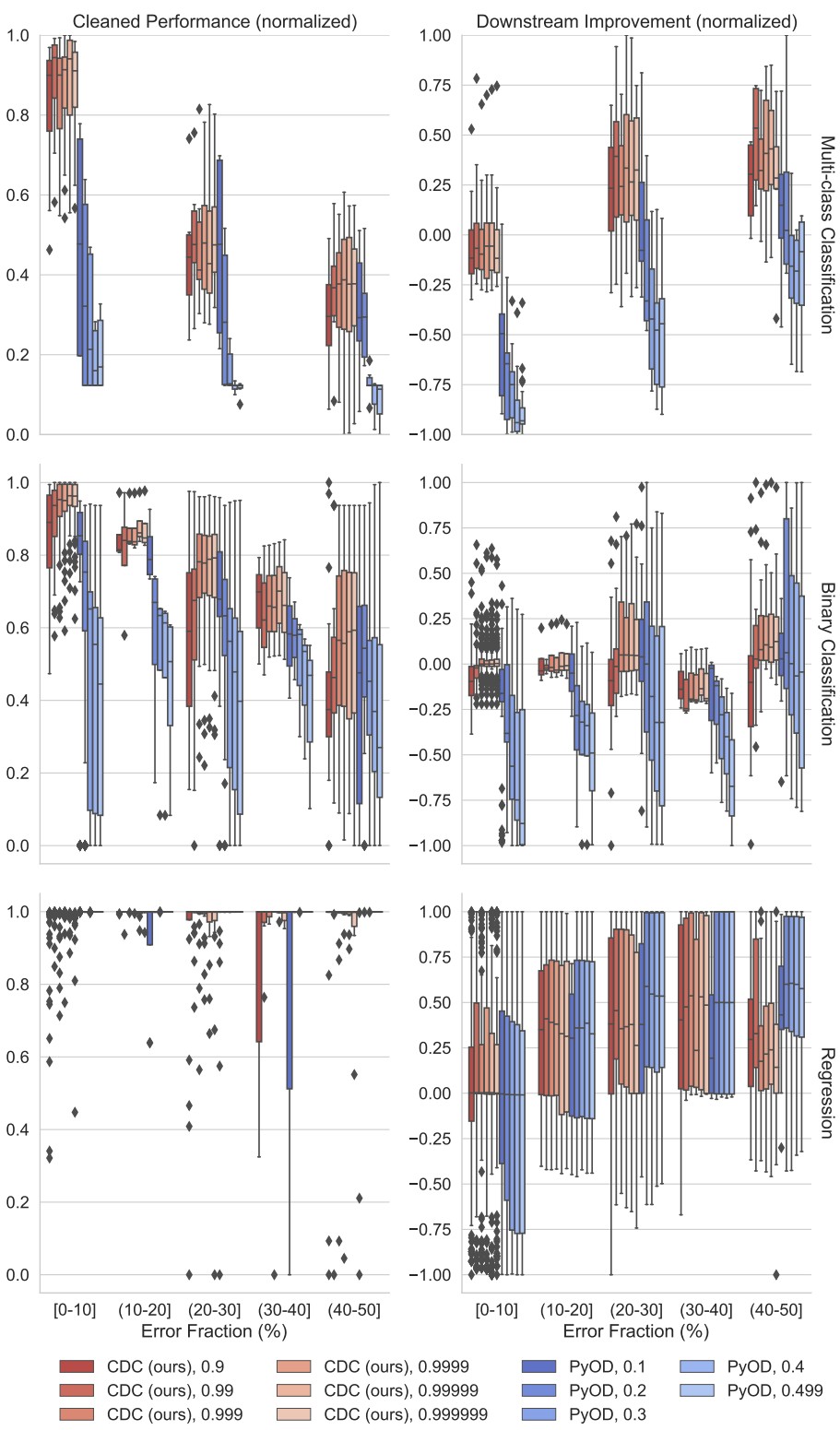

Figure 5: (*Left*) Cleaned performance, (*Right*) downstream improvement (normalized), and rows represent downstream tasks. For regression tasks, the downstream improvements are more dispersed, less pronounced, and with larger error fractions worse than the baseline.

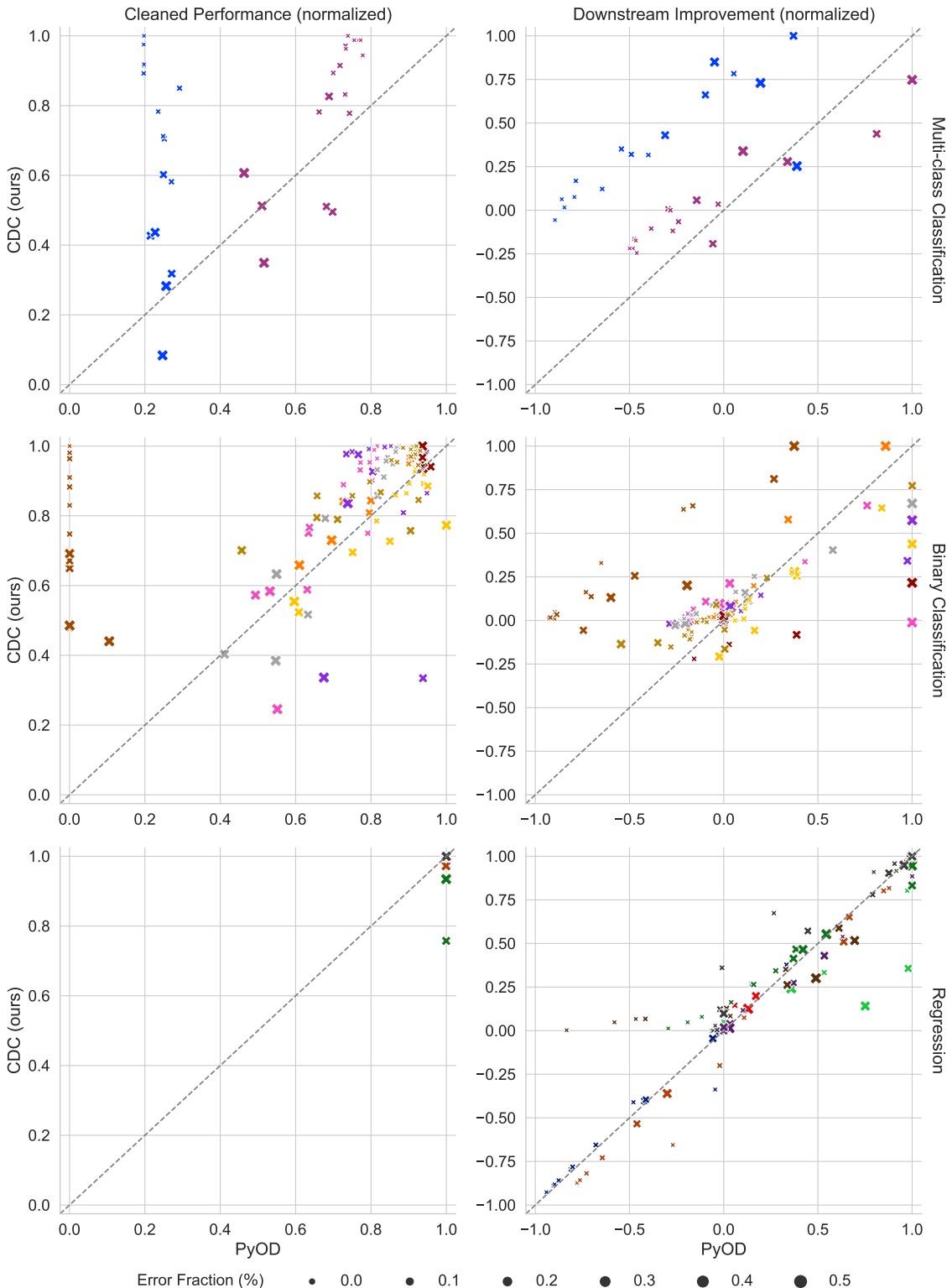

Figure 6: (*Left*) CDC vs. PyOD baseline data cleaning performance, (*Right*) improvement in predictive performance in the downstream task (normalized), and rows represent downstream tasks. We observe empirically that the improvement with our proposed method appears to be less pronounced for downstream regression tasks.

