# OpenReview forum: "Conformal Data Cleaning: Statistical Guarantees for Data Quality Automation in Tables"
_TMLR — Rejected by TMLR_

### Review · Reviewer_rF5C · 2023-04-18

**Summary Of Contributions:**

The authors propose a novel cleaning approach by combining an application-agnostic ML-based data cleaning method with conformal prediction. The method is evaluated on multiple tabular datasets and against a state-of-the-art baseline, i.e., data cleaning on the PyOD approach (Zhao et al., 2019), one of the best approaches for detecting outliers. The experimental results show that the proposed approach achieves higher cleaning performance and improves the performance of downstream tasks (i.e., classification, regression).

**Audience:**

Yes

**Broader Impact Concerns:**

--

**Claims And Evidence:**

Yes

**Requested Changes:**

The authors should check/evaluate whether more recent tabular data generation approaches (see references provided above) used on top of PyOD can make up for a better baseline.

**Strengths And Weaknesses:**

Strengths:
1. The paper provides a solution that can be integrated into data analytics pipelines for tabular data to automate data cleaning processes.
2. The proposed approach inherits some desired properties and conformity guarantees from conformal prediction.
3. The paper is well-written and the approach well-presented.
4. The topic of the paper is important for many real-world applications using tabular data.

Weaknesses:
1. The proposed solution combines well-known techniques.
2. There are also more advanced approaches for data cleaning which could be used on top of the PyOD approach (Zhao et al., 2019), see for example Borisov et al. (2022). Language models are realistic tabular data generators. arXiv preprint arXiv:2210.06280.
Other examples are:
Zhang, Yishuo, et al. "GANBLR: a tabular data generation model." 2021 IEEE International Conference on Data Mining (ICDM). IEEE, 2021.
Rajabi, Amirarsalan, and Ozlem Ozmen Garibay. "Tabfairgan: Fair tabular data generation with generative adversarial networks." Machine Learning and Knowledge Extraction 4.2 (2022): 488-501.
Xu, Lei, et al. "Modeling tabular data using conditional gan." Advances in Neural Information Processing Systems 32 (2019).

---

> ### Author Response · Authors · 2023-04-26
>
> Thank you very much for your valuable feedback. All the mentioned papers have in common that they use generative approaches to synthesize tabular datasets. You are completely right that these methods could be helpful, especially in addition to or as a replacement for the PyOD models used. We believe that in empirical comparisons it is easier to disentangle the effects of the conformal prediction-based outlier detection (Figure 2 in the paper) and imputation methods, for which simple discriminative models were demonstrated to perform slightly better than generative models (Jäger et al. (2021)). In fact, we already plan to similarly use generative models for data cleaning.
>
> We added some text (highlighted in blue) in Section 4.2, Section 6.4, and Chapter 7 to make this more clear.

---

### Review · Reviewer_V7RC · 2023-04-30

**Summary Of Contributions:**

The authors propose an automatic data cleaning framework based on conformal regression and classification. Essentially, the confidence intervals and sets are used to determine whether a cell in a tabular dataset is an outlier or not, aber on a model trained to predict this column from the others. If so, the data is imputed by the model prediction. If the confidence set is empty, nothing is done.

**Audience:**

Yes

**Broader Impact Concerns:**

No concerns.

**Claims And Evidence:**

No

**Requested Changes:**

See above.

**Strengths And Weaknesses:**

Strengths:
- Promising combination of approaches - having data cleaning with guarantees is definitely interesting.
- Good background and related work.
- Thorough experiments and detailed description of experiments.
- Encouraging results in the sense that the approach improves over baselines.

Weaknesses:
- I am not convinced that the authors' interpretation of the conformal guarantee is correct. Specifically after Equation (6). Essentially the coverage guarantee only holds on inliers (assuming the model was trained and calibrated on clean data). Conformal prediction does not allow to say anything about outliers unless specific methods for conformal prediction with distribution shift are used. That is, the conclusion that the column value is not in the confidence set for an outlier with at least peobability 1 - alpha does not hold as of my understanding. Even using conformal methods for outlier detection, guarantees only hold for inliers (inliers are correctly detected as inliers with probability 1 - alpha), not for outliers. This is a critical flaw of the approach in my opinion.
- The treatment of empty confidence sets is also not meaningful as outliers might as well get empty confidence sets (as we cannot say anything about outliers, see above).
- In the experiments it is unclear how the calibration set is chosen - how large is it, does it contain errors, etc? I assume the training set does not contain errors in which case I would assume clean training and calibration sets to be rather small in practice. Experiments with smaller numbers of training and calibration sets would be interesting.
- I am also missing whether there are random trials of calibration/test slit performed?
- It would also be great to see coverage and confidence set sizes for individual columns as examples, vs error rate.
- A distinction between performance on regression and classification  would also be interesting.
- In terms of writing the paper is very verbose and could be shortened a bit in my opinion. the algorithm is not too insightful and while I appreciate providing a nice API, I would defer this to the code documentation or appendix. Background is also longer than required at times.

Conclusion:

I like the problem and idea but am not convinced that conformal prediction has been applied correctly in this setting. As a result, the experiments are purely empirical and using conformal prediction does not have advantages. If this could be fixed and there would eg be a guarantee on inliers or so, I believe it would be much more compelling. Experiments would also have to be adapted to give insights into the conformal prediction bits (coverage, set sizes by error rates, calibration/test trials etc.). In it's current form I believe the paper should not be published at TMLR.

---

> ### Author Response · Authors · 2023-05-24
> **Answers to the comments regarding weaknesses**
>
> We thank the author for their valuable review.
>
>
> > I am not convinced that the authors' interpretation of the conformal guarantee is correct. Specifically after Equation (6). Essentially the coverage guarantee only holds on inliers (assuming the model was trained and calibrated on clean data). ( ... ) Even using conformal methods for outlier detection, guarantees only hold for inliers (inliers are correctly detected as inliers with probability 1 - alpha), not for outliers. This is a critical flaw of the approach in my opinion.
>
> You are right the conformal framework does not give guarantees about outliers but, as you mentioned, about inliers. Our formulation was incorrect and misleading - thanks a lot for catching this mistake (after Equation (6)). We reformulated the text and highlighted this critical detail. In our cleaning heuristic, we test whether a cell conforms well with the training data, and if so, no data cleaning should be performed, as the data points are considered inliers. Otherwise, we assume the cell's value is erroneous (outlier) and try to fix it. In the revised version of the manuscript, we emphasize this assumption that the data which cannot be considered inliers with statistical guarantees are outliers. Besides the theoretical guarantees, a major focus of our work was the complementary empirical evaluation, which demonstrates that the proposed heuristic is effective in data cleaning applications. We changed some phrases in Sections 1 and 3.2 (highlighted in green) to emphasize this.
>
> > The treatment of empty confidence sets is also not meaningful as outliers might as well get empty confidence sets (as we cannot say anything about outliers, see above).
>
> The empty set treatment is empirically motivated since we recognized that distribution shift results in detecting a large number of erroneous cells. Cleaning them resulted in even worse downstream performance than not cleaning at all. We extended the paragraph in Section 4.1 to explain this more clearly. This empirical finding is not backed by theoretical guarantees, but in extensive empirical evaluations, we find that this approach does improve the downstream performance of ML models after cleaning. We would like to emphasize the difficulties with data shifts in general - most approaches in Machine Learning assume stationary data. So even if there were guarantees for the conformal prediction framework, the downstream models would likely also become impacted. We highlight these limitations in the revised manuscript in Section 4.1
>
> > In the experiments it is unclear how the calibration set is chosen - how large is it, does it contain errors, etc? I assume the training set does not contain errors in which case I would assume clean training and calibration sets to be rather small in practice. Experiments with smaller numbers of training and calibration sets would be interesting.
> I am also missing whether there are random trials of calibration/test slit performed?
>
> See below.
>
> > I am also missing whether there are random trials of calibration/test slit performed?
>
> For the experiments, we use 1000 (clean) calibration data points, following recommendations from 1, and repeat them three times, which also resamples the calibration set. To highlight these points and the entire experimental setup more clearly, we have rewritten Section 4.4 (highlighted in green). The general trends remain the same, however, there are some differences between classification and regression tasks. The performance gap is less pronounced for regression and with larger error fractions the baseline can outperform CDC.
>
> > It would also be great to see coverage and confidence set sizes for individual columns as examples, vs error rate.
>
> We agree that metrics for the conformal framework would help to understand CDC in more detail and potentially reveal limitations. Unfortunately, this needed some larger adaptions on the code base and forced us to rerun the experiments (currently running). To not delay the reviewing process, we will include the metrics on coverage and confidence set sizes in the supplementary material as soon as they are ready.
>
> > A distinction between performance on regression and classification would also be interesting.
>
> We added Appendix C (highlighted in green), which shows and discusses the experimental results for each of the downstream tasks separately.
>
> > In terms of writing the paper is very verbose and could be shortened a bit in my opinion. the algorithm is not too insightful and while I appreciate providing a nice API, I would defer this to the code documentation or appendix. Background is also longer than required at times.
>
> We agree that some paragraphs could be shortened to focus more on the relevant contributions. In the revised version of the manuscript, we included several minor changes to sharpen the focus of the manuscript. As suggested, we moved the API documentation to the Appendix to shorten the manuscript. Thanks for that suggestion!

---

### Review · Reviewer_mXky · 2023-06-05

**Summary Of Contributions:**

Authors apply the idea of conformal prediction to the task of cleaning tabular data.  First the conformal data cleaning (CDC) trains a classifier to detect statistical outliers.  Next, rather than a (0,1) prediction, this is replaced by a (N/A, pred) where the pred is the output from the classifier.  Effectively, examples with N/A have no change, whereas examples that have a "pred" have their values replaced, and are thereby cleaned.  This is applied to 18 datasets and compared against PyOD data cleaning method.  In about 75% of cases, using CDC leads to an improvement in downstream performance.

**Audience:**

Yes

**Claims And Evidence:**

Yes

**Requested Changes:**

The authors already suggested a number of improvements:
  - Working on optimizing tree-based models
  - Trying generative NN methods

Other options include testing against more baselines as well:
  - Adaptation Layer (Goldberger and Ben-Reuven) https://openreview.net/forum?id=H12GRgcxg
 - Clean or Annotate (Chen et al., https://arxiv.org/abs/2110.08355) offers 4 methods of data cleaning: AUM, Cartography, Large Loss, Prototype, as well as a number of data denoising techniques to consider.
  - Loss Correction (Patrini et al. https://ieeexplore.ieee.org/document/8099723)
  - Confident Learning (Northcutt et al. https://arxiv.org/abs/1911.00068)
  - Using LLMs (Chong et al. https://aclanthology.org/2022.emnlp-main.618/)
The above papers should all be cited as well and discussed in related work.  ( Some are already discussed, but others are missing)

Overall: The results under the proposed method don't seem strong enough to outperform existing techniques.


**Strengths And Weaknesses:**

Strengths: Tabular data cleaning is a incredibly important problem with many practical applications.  Almost every company in the world has spreadsheet data somewhere and more often than not, this data is noisy.
   - CDC often leads to improvements in quality and downstream performance
   - CDC is statistically sound and many details of the algorithm were included (maybe even too much, some could be moved to appendix)
   - The paper is reasonably well written and the experiments are well structured with 3 trial runs across many settings and datasets.

Weaknesses: The technique just doesn't seem to work that well, even given all the advantages of a nice clean setup.
   - The training set is assumed to be clean, which is often not true in real life
   - The method gets access to thousands of cells for training a cleaner, which again is often unrealistic
   - CDC does not seem to help at all in a quarter of the cases
   - The technique does not apply to text or image type data
   - The process is slow and a new model must be trained for every column
   - The authors only compare against one baseline (PyOD), when there are many data cleaning and outlier options available
   - Even compared to this one baseline, CDC has poor performance in TPR
   - While CDC is better than doing nothing, it is really hard to tell (ie. based on Fig 4) that it does better than other alternatives.

---

> ### Author Response · Authors · 2023-06-08
> **Answers to Review**
>
> We thank the authors for their feedback.
>
> > * The training set is assumed to be clean, which is often not true in real life
> > * The method gets access to thousands of cells for training a cleaner, which again is often unrealistic
>
> We agree that the assumption of clean training data is a strong one. Yet availability of clean training data is also one of the fundamental assumptions of training most machine learning models, as most approaches do not explicitly model the noise distribution on a given data set. Assuming that the data errors only happen at inference time is also motivated by real-world use cases, for instance when ML models are deployed to production systems: At training time and during model development, there is more time for data preparation, including manual cleaning efforts.
> Given that, we believe it is an advantage of our approach that we can use the entire training dataset to train the cleaner without further labelling/data collection/rule mining effort. At test/inference time however, a higher degree of automation for data cleaning procedures is desirable.
>
> > * CDC does not seem to help at all in a quarter of the cases
>
> In about 67% of our experiment settings, at least one HP of CDC (which we tested) improves the downstream performance. The baseline achieves this in only 43%, which is, in our opinion, a substantial improvement given the high degree of automation CDC offers. We added some text in Section 5.2 (highlighted in red) to state this more clearly.
>
> > * The technique does not apply to text or image type data
>
> Indeed the focus on tabular data can be considered a limitation. And we did consider other modalities in our earlier work on data cleaning. This often came at the price of less controlled experiments - when dealing with complex data preparation pipelines and a variety of downstream task models, there are many factors that render fair comparisons more difficult. Focusing on tabular data allowed to consider a large number of data sets and experimental conditions. Besides we agree with your comment that tabular data is (and probably will remain) an important modality with many practical applications. In the revised version of the manuscript we highlight this limitation and the reasons behind our experimental setting.
>
> > * The process is slow and a new model must be trained for every column
>
> It is true that CDC needs to fit one model for each column, as it is based on established imputation approaches. However, because the cleaning models are relatively simple and can run in parallel, the computational overhead is not too large in terms of wall-clock runtime.
>
> > * The authors only compare against one baseline (PyOD), when there are many data cleaning and outlier options available
>
> One of the ideas behind our approach is that we do not propose a specific model for cleaning and do not claim to outperform a specific cleaning model. Instead we propose a procedure that can be combined with any imputation or outlier detection model (added a sentence in Section 1 (red) to make this more clear). For the imputation model we opted for an AutoML solution, which underlines the model agnostic nature of our proposed approach. Similarly, we chose PyOD as baseline because it integrates 40 different outlier options exposed through a unified API. For our experiments, we then simply used the outlier detector in PyOD that has been demonstrated to perform best. As highlighted in Section 2, for the sake of comparability and because we aimed at a high degree of automation in the data cleaning process we did not include cell-based cleaning approaches that rely on user input.
>
> > * Even compared to this one baseline, CDC has poor performance in TPR
>
> Since the other results show that CDC outperforms the baseline, other factors are also important for good results, e.g., smaller FPR and accurate cleaning. We added some text (highlighted in red) to state this clearly Section 6.1.

---

> ### Author Response · Authors · 2023-06-08
> **Answers to Review (cont.)**
>
> > * While CDC is better than doing nothing, it is really hard to tell (ie. based on Fig 4) that it does better than other alternatives.
>
> We added a few sentences in Section 5.2, 5.3, and Appendix C (highlighted in red) to emphasize that CDC does not only perform better than the baseline but also leads too downstream improvements more often.
>
> > * Working on optimizing tree-based models
>
> During experiments, we also tested XGBoost and LightGBM, which were both similarly good as other tree-based approaches but much slower. For this reason, we decided to remove them and stick to the "classic" tree models.
>
> > * Trying generative NN methods
>
> Thanks for this suggestion, we also received this comment from reviewer rF5C, see also our comment here <https://openreview.net/forum?id=XFWEvmEyBp&noteId=dpbAeykkNR> and/or in the revised manuscript (highlighted in blue) in Section 4.2, Section 6.4, and Chapter 7.
> In general generative methods have been demonstrated to perform not as well as discriminative methods in a fully conditional imputation setting, which is why we opted for the state of the art in imputation methods for this study.
>
> > The above papers should all be cited as well and discussed in related work.
>
> Thank you very much for these pointers. We include and discuss them in Section 2 (highlighted in red).
>
> > Overall: The results under the proposed method don't seem strong enough to outperform existing techniques.
>
> In our study we focus on improving the degree of automation in data cleaning, while being model agnostic. It is not uncommon in empirical studies that methodological or empirical contributions do yield improvements in all cases. We are convinced that comprehensive evaluations with the appropriate empirical rigour, including conditions that are realistic and challenge the proposed solution, are important. In our study we tested the proposed cleaning procedure in a wide spectrum of tasks, data sets and error conditions. In the majority of cases we find that the proposed approach helps while offering a higher degree of automation than most other cleaning approaches.

---

### Decision · Action_Editors · 2023-06-29

**Recommendation:** Reject

**Comment:**

While an interesting idea, the reviewers are all leaning towards rejection as they believe that the current usage of the conformal prediction framework has not been performed appropriately. All reviewers suggested more baselines and metrics that would benefit this work in a potential future version.

**Audience:**

The reviewers all highlighted that this is an interesting idea, however its execution and the obtained results limit the applicability of the method.

**Claims And Evidence:**

The reviewers have highlighted a mismatch between the conformal prediction framework and how the authors are using it in this work. The reviewers have also pointed out that evaluation metrics referring to the conformal prediction framework were missing, and that the proposed baselines were weak.

**Resubmission Of Major Revision:**

The authors may consider submitting a major revision at a later time.